

# Combining fire radiative power observations with the fire weather index improves the estimation of fire emissions

Francesca Di Giuseppe[1], Samuel Rémy[2], Florian Pappenberger[1], and Fredrik Wetterhall[1]

[1]European Centre for Medium-Range Weather Forecasts (ECMWF), Reading, UK
[1]Institut Pierre-Simon Laplace (IPSL), CNRS/UPMC, Paris, France

*Correspondence to:* Francesca Di Giuseppe (francesca.digiuseppe@ecmwf.int)

**Abstract.** The atmospheric composition analysis and forecast for the European Copernicus Atmosphere Monitoring Services (CAMS) relies on biomass burning fire emission estimates from the Global Fire Assimilation System (GFAS). GFAS converts fire radiative power (FRP) observations from MODIS satellites into smoke constituents. Missing observations are filled in using persistence where observed FRP from the previous day are progressed in time until a new observation is recorded. One of the consequences of this assumption is an overestimation of fire duration, which in turn translates into an overestimation of emissions from fires. In this study persistence is replaced by modeled predictions using the Canadian Fire Weather Index (FWI), which describes how atmospheric conditions affect the vegetation moisture content and ultimately fire duration. The skill in predicting emissions from biomass burning is improved with the new technique, which indicates that using an FWI-based model to infer emissions from FRP is better than persistence when observations are not available.

## 1 Introduction

Emissions generated from biomass burning contribute extensively to the global budget of several atmospheric constituents such as aerosols, carbon monoxide (CO) and dioxide ($CO_2$). The annual emission for several gases generated during combustion is comparable to what is emitted from anthropogenic sources (Crutzen et al., 1979). Chemical fluxes from fires need to be accounted for in global systems that monitor and forecast atmospheric composition, such as the European Copernicus Atmosphere Monitoring Services (CAMS). Since 2012, the Global Fire Assimilation System (GFAS) [1] provides fire emissions to the atmospheric composition model operated by CAMS: the Integrated Forecasting System (IFS). GFAS uses a liner relationship to convert Fire Radiative Power (FRP) observations from the MODIS instruments (Kaufman et al., 2003) into fuel consumption. Emissions are then derived using land-use dependent conversion factors. The current system does not explicitly forecast fire evolution.

---

[1]The Global Fire Assimilation System (GFAS) has been developed initially under two EU founded projects MACC and MACC-II. Since 2015, it is operated as part of the European Copernicus Atmosphere Monitoring Services (CAMS)





The main shortcoming of the MODIS FRP product is the limited sampling frequency: between 1 and 4 observations per day and per satellite for each GFAS grid point (Kaiser et al., 2012). The typical coverage has a return period between 4 and 12 hours for any location but this time window is often expanded by the presence of clouds, snow and ice on the ground, which interfere with the instrument detection systems (Kaufman et al., 2003). When a missing observation occurs the day after a fire has been detected, the GFAS relies on persistence. This means that values of FRP for the previous day as provided by GFAS are progressed in time until a valid observation is detected. The assumption of persistence can potentially cause an overestimation of fire duration, and this will be more pronounced in areas of the planet that are frequently covered in clouds or where the time variability of fires is the greatest. To put things into context, the June-August 2013 observed global average fire duration calculated using MODIS observations was 4.2 days. By assuming persistence, GFAS extends this duration by 1.2 days (see figure 1 in the results section).

Increasing the temporal frequency of observations could reduce the need to resort to the persistence assumption. In 2008, the EUMETSAT Land Satellite Application Facility started the real-time production of a newly developed FRP product generated from SEVIRI observations. This product maintains SEVIRI's return (sampling) period of 15 minutes, which makes it capable of resolving the diurnal cycle of open fires in Africa and southern Europe with unprecedented accuracy. The high sampling frequency is also an advantage for fire observations during brief cloud-free spells in otherwise mostly cloudy regions. In addition it covers higher latitudes than the MODIS fire products since snow and ice pixels are being processed. Despite these advantages the FRP product from SEVIRI does not provide global coverage since it only covers a geographical disc comprising Europe and Africa. Moreover, the SEVIRI FRP level 2 products undergo a different calibration procedure from that of MODIS, and the merging of the two datasets has proven to be a challenge (Roberts et al., 2011).

Given the difficulties in the use of multiple platform observations, in this paper we propose a new approach to improve the emission estimations from FRP observations based on the idea that weather plays a major role in the ignition, spread and termination of wild fires wherever there is available combustible vegetation and suitable terrain topography (Flannigan et al., 2005, 2009). The use of a fire danger index based on atmospheric parameters, such as the Fire Weather Index (FWI)(Van Wagner, 1985), is potentially a better approach to simulate the evolution of fires compared to the persistence assumption (De Groot et al., 2007). The FWI is already widely employed in fire management and control (Lee et al., 2002). However, it does not explicitly model fire evolution but a measure of fire danger (Van Wagner et al., 1987). Even for extreme FWI values there is a need for a stochastic component, i.e. an ignition, to start a fire. For this reason, situations in which FWI is high but no fire is recorded are not uncommon. (Di Giuseppe et al., 2016) showed how this index can be a very good predictor of fire activity where fire is limited by moisture (e. g. boreal forests), whereas where fires are fuel limited (e.g. in the savanna ecosystem) the stochastic component becomes dominant. In these regions using the FWI as a fire modulator could be less accurate. Still, in the proposed application FWI is used when a fire has already been detected by a valid value of FRP, therefore the random component (aka ignition) is intrinsically removed. In this situation the FWI can be used to predict how atmospheric temperature, humidity, precipitation and wind affect the fuel moisture content and its inflammability (Van Wagner, 1985; Van Wagner et al., 1987; Stocks et al., 1989). Moreover on the wave of the same idea, Di Giuseppe et al. (2017) showed how the CAMS





emissions forecast can be improved if fire emissions are not kept constant during the forecast integration but predicted by FWI evolution.

The new model based on the FWI has been implemented in GFAS. Improvements in the prediction of biomass-burning emissions have been assessed for a collection of 5-day forecasts initialized on a daily basis during three months spanning
June-August 2013.

## 2  Method

Fire Radiative Power is the radiant heat power emitted by fires and it can be related to the energy released during the combustion process, whereby carbon-based fuel is oxidized to $CO_2$ (Wooster et al., 2005). Measurements of FRP are proportional to the total biomass combustion and thus to emissions (van der Werf et al., 2006; Kaiser et al., 2012).

The FRP ($\rho$) observations from MODIS used in this study are available at 1 km resolution. In the GFAS these observations are first area-weighted by the portion of satellite footprint included in the grid box area in order to provide mean FRP density on a grid box of 0.1 degree. They are then converted into dry matter burnt $\bar{\rho}$ following (Wooster et al., 2005) and finally into emissions for 44 constituents, $\mathcal{E}_i$, using the simple formulation

$$\mathcal{E}_i = \bar{\rho} \cdot \mathcal{X}_i \tag{1}$$

where $\mathcal{X}_i$ are conversion coefficients from Heil et al. (2010).

Since observations contain gaps (mostly because of cloud cover) a simple model is applied to obtain a practical estimate of current observations from earlier measurements (Kalnay, 2003). At a specific time step t the estimate for $\rho_t$, $\widetilde{\rho}_t$, given observation at t-1, is then expressed as

$$\widetilde{\rho}_t = O_{t-1 \rightarrow t} \cdot \rho_{t-1} + \delta\rho \tag{2}$$

where $\delta\rho$ is the observed FRP increment in the $\delta t$ time interval.

$$\delta\rho = \rho_t - \rho_{t-1} \tag{3}$$

and $O$ is an "observation operator" that transfers the information at time t-1 to time t. As already explained, persistence is assumed in the operational version of GFAS and the observation operator is equal to the identity operator, $O \equiv I$ (Kaiser et al., 2012). Therefore, in the case of a non-valid observation at time t, $\delta\rho$ becomes undefined and equation 2 simply reduces to
$\widetilde{\rho}_t = \rho_{t-1}$. When a valid observation is available, equation 2 leads to $\widetilde{\rho}_t = \rho_t$.

We propose the use of a new formulation for the observation operator $O$ which relies on the actual weather conditions and the fire conditions recorded the previous day (French et al., 2004). The FWI provides the base for $O$. FWI is defined as the intensity of the propagating flame front depending on the quantity of energy released from a linear unit of the front itself. It is an index composed of six sub-indices referring respectively to the daily variation of water content for fuels with different





response times to changes in weather conditions, the initial rate of spread for propagation, the quantity of fuel and the expected intensity of the flame front. More details on the FWI model are provided in appendix A1 while full details can be found in (Van Wagner, 1985).

To take into account uncertainties in the weather parameters, every day an ensemble of FWI values is diagnosed from the 51 weather forecast realizations of the European Centre for Medium-Range Weather Forecasts (ECMWF) ensemble forecasting system (Buizza et al., 1999) and the ensemble mean is evaluated. Different ranges of values can correspond to different fire danger conditions in different regions. A local calibration is performed to convert the absolute value of FWI into a normalized index. This is achieved by employing historical values of FWI calculated using ECMWF reanalysis, ERA-Interim, as forcing for the period 1980-2014 (see Di Giuseppe et al. (2016) for details) and calculating for each location the FWI cumulative distribution function (CDF) and its inverse. The observation operator $O_{t-1 \to t}$ is then defined as

$$O_{t-1 \to t} = CDF^{-1}(FWI(t-1)) \tag{4}$$

$O$ is therefore a direct function of temperature, relative humidity, precipitation and wind through the FWI formulation. For any value of FWI at time t-1, $O$ provides a normalized index with values in the range $[0; 1]$ which is then multiplied by the previous day observed FRP measurements $\rho_{t-1}$, to provide the best FRP estimation, $\widetilde{\rho_t}$, from which biomass-burning emissions are then calculated from equation 1.

## 3 Results

To test the impact of the new observation operator on GFAS fire emissions and on the biomass-burning aerosol plumes predicted by the Integrated Forecasting System (IFS) operated by CAMS, three months of simulations were performed for summer 2013. In total 90 days of new fire emission data were produced and used in the IFS (Flemming et al., 2015). For each of the 90 starting dates the IFS produced a 5-day forecast of a number of atmospheric constituents. In this paper, we focus on the impact of the new fire emissions on biomass-burning aerosols.

### 3.1 Consequences of the persistence assumption

Figure 1a demonstrates the more obvious consequence of the persistence assumption; the increase in mean fire duration. It shows the difference in fire duration between MODIS and GFAS. The density plot takes into account all fires detected by MODIS instruments on board the Terra and Aqua platforms during the verification period in summer 2013. Fire duration for MODIS is defined by the number of continuous days for which FRP is available and larger than zero. Fire duration in GFAS extends also during observation black-out periods until a valid observation confirms the end of the event. The observed global average fire duration in the MODIS dataset is 4.2 days, and by assuming persistence GFAS increases the fire duration by 1.2 days. In the new formulation $O$ takes on values in the interval $[0; 1]$, and FRP predicted by the new model is therefore by construction expected to be smaller than, or equal to what is predicted by persistence. This new formulation means an overall





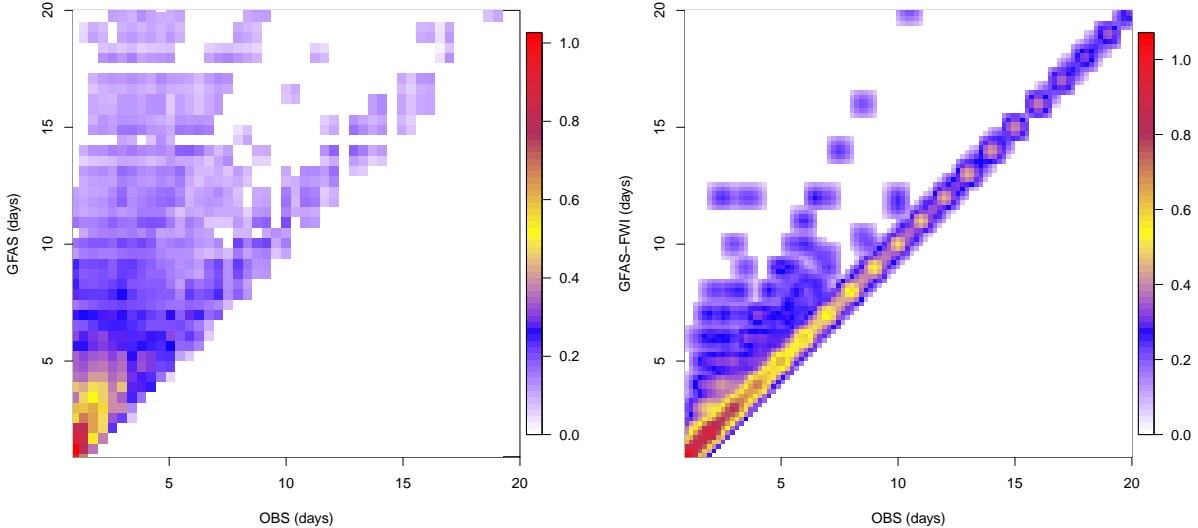

**Figure 1.** Left panel: normalized density plot of average consecutive days of continuous fire observations as recorded by MODIS on board of Aqua and Terra satellite versus the estimated values from the GFAS assimilation system. The longer fire duration in GFAS is due to the persistence assumption, which extends observations during observation black-out periods. Right panel: as before but for the application of the new fwi based model.

decrease of the mean fire duration (1b). With the new method the mean daily duration of fires is still large with GFAS than with MODIS, but the overestimation is reduced to 0.3/days on average.

As the assumption of persistence produces a systematic increase in fire duration, the emissions from biomass-burning are also likely to be overestimated. Unfortunately, the latter assumption is difficult to verify as only limited information is available on the real amount of mass fluxes released into the atmosphere during fires. Ground truth measurements for fire emissions are only available from controlled ignition experiments aiming at studying the combustion process in detail (e.g the TROFEE campaign described in Karl et al. (2007); Yokelson et al. (2008)). These experiments are conducted on isolated spots and are not available at the global scale needed to validate the emissions calculated by the GFAS. Global coverage of biomass burning products can be provided by sensors on board polar orbiting or geostationary satellites. However all sources of emission, including for example anthropocentric fuel burning and chemical reactions, contribute to these measurements, not only biomass combustion. Disentangling the sole contribution from fire emissions remains a challenge. Still, a strong hint that the persistence assumption is responsible for an excess overestimate of emission is provided by figure 2 where the averaged carbon monoxide (CO) concentration at surface observed from the MOPITT [2] instrument on board the Terra satellite (Deeter et al., 2003) are compared with analysis by the IFS operated by CAMS. While the most relevant source of CO is natural in origin due to photo-

---

[2]Data used in the analysis are version 6 Level 3 TIR/NIR products combining daytime and nighttime observations. Details of the retrieval algorithm can be found in Deeter et al. (2014). Data are available at https://eosweb.larc.nasa.gov/project/mopitt.

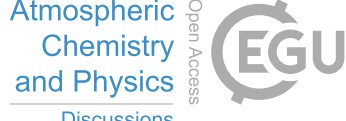


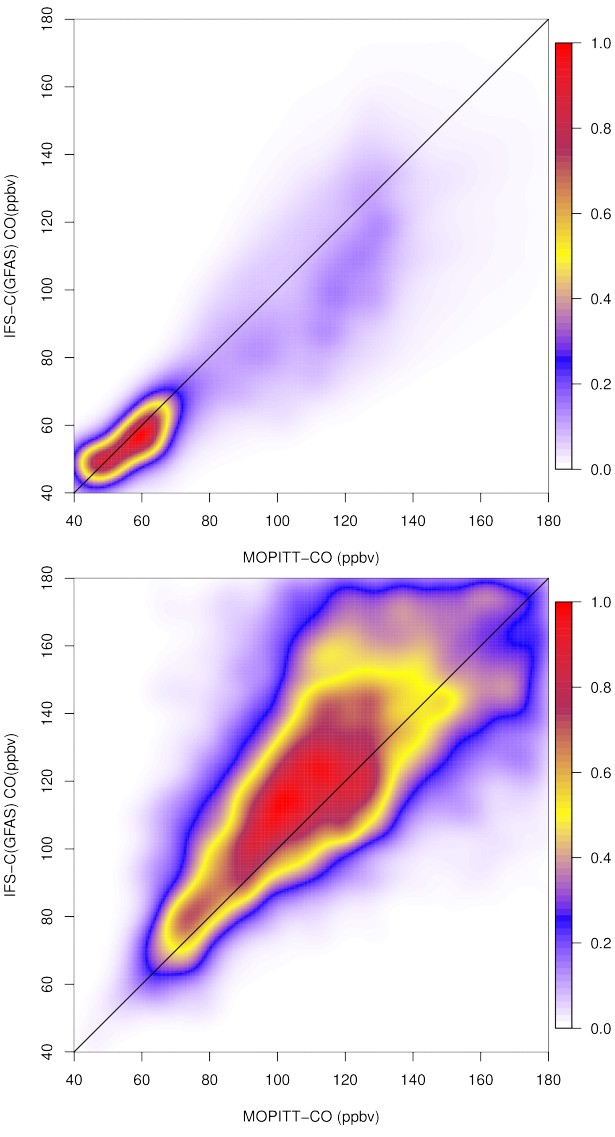

**Figure 2.** Normalized density plot of monthly mean surface concentration for 2013 from the MOPITT instrument on board the Terra satellite plotted against CO surface concentration analysed by CAMS. Points are grouped in locations for which fires were not detected by MODIS (upper panel) and locations where fire activity was observed during that month (lower panel).

chemical reactions in the troposphere, a large contribution also arises from combustion processes, including volcanoes and fires (Kasischke and Bruhwiler, 2002). CO can therefore to some extent be used as a marker for fire activities. Comparing CO concentration for all locations for which no fire events were recorded by MODIS (i.e. FRP ==0, Fig. 2a), with locations where fire activity was detected (Fig. 2b) it is evident the increase in CO surface concentration during fire combustion. This increase
5   is correctly picked up by the CAMS analysis which is initialized with an extra source of CO provided by the GFAS emissions.





The important aspect of Figure 2 is that when the CO source from fire emission is included the CO surface concentration is overestimated by the CAMS analysis. The mean bias (CAMS minus MOPITT) is slightly negative (-5 ppbv) when fire is not observed and becomes highly positive (+95 ppbv) in case of fire events. Although transport and sinks could also play a role in this large overestimation, CO emissions from GFAS are probably also overestimated.

## 3.2   Model behavior

To understand the impact of the new operator on the emissions we concentrate on the prediction of extreme fire. In July 2013 large fires were recorded around the Hudson bay in the Quebec province, Canada. Hundreds of fires consumed a total area of 616,000 hectares during the first two weeks of July 2013. Two major events occurred, the first on 2-6 of July with peak activity on the 4 June (figure 3) and the second between 8-10 of July. The smoke plumes produced by the fires in Canada reached Europe on 12 July 2013.

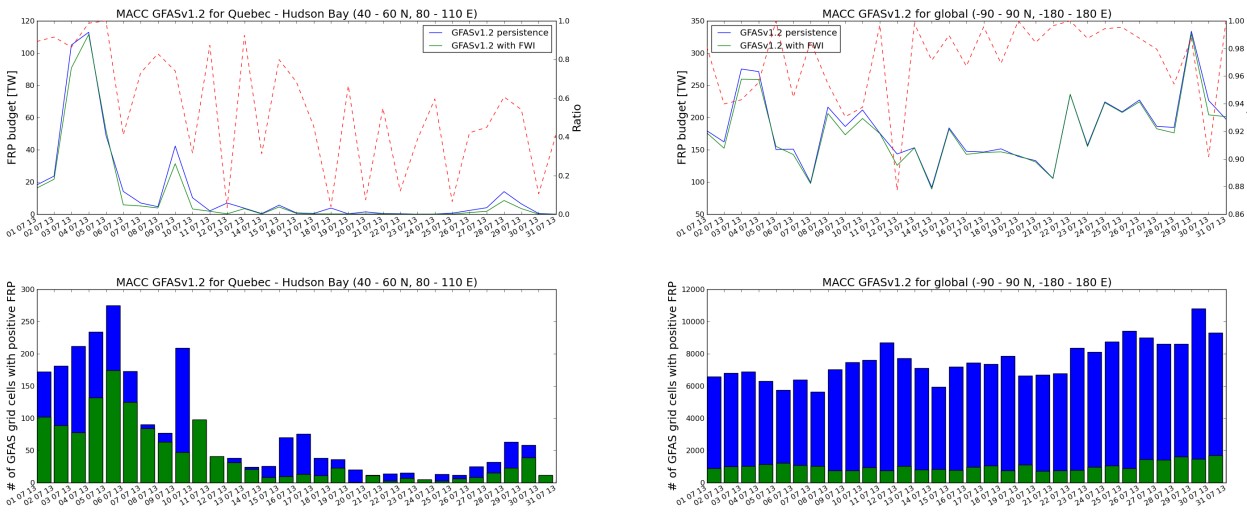

**Figure 3.** Budget of FRP over the Hudson Bay area and global (top panels). GFAS using persistence as an observation operator is shown in blue, using FWI as an observation operator in green. The ratio between the two experiments is indicated by the dashed red line. The histogram (bottom panels) shows the number of GFAS grid cells with positive observations for every day in July over the selected areas (blue bars) and among these the number of grid cells for which observational data was not available and FRP was estimated using either the persistence or the FWI based model (green bar).

Figure 3 show a comparison of FRP budgets over the Hudson Bay area (40-60N, 70-100W) provided by the GFAS using the persistence hypothesis and FWI respectively. Using FWI reduced the total FRP by up to 80% within the Hudson Bay area. The histogram also highlights the number of grid cells for which a positive FRP was estimated, and the number of grid cells for which observational data was not available, i.e.. the output of the GFAS was computed using persistence or FWI. It is interesting to note that the proportion of the GFAS grid cells without observational data increases as the fire event progresses





in time, on 5-7 July. On a global scale, in around 20% of cases, FRP from the GFAS is estimated either assuming persistence or using the FWI model.

The global budget shows a reduction of FRP computed using FWI in the range of 0 to 5 % generally, up to nearly 10 % at times. The difference varies markedly from region to region, for example the FRP budget over Africa north of the Equator is not affected at all by this modification (not shown). This is explained by the fact that the weather over this area is mostly dry during the summer, which means that $O$ is very close to 1 and it is therefore equal to assuming persistence.

**Figure 4.** On the left column; aerosol optical depth (AOD) at 550nm for the IFS forecast starting on 10/7/2013 and with lead times of 3 hours (panel a), 24 hours (panel b) and 48 hours (panel c) using the operational GFAS with observation operator equal to persistence. On the right column difference in AOD between the two experiments using the classical GFAS-persistence and the GFAS-FWI observation operator. Panel d; 3 hours , panel e; 24 hours , panel f; 48hours.



The reduction in FRP using the new operator in GFAS had a significant impact on the simulations of biomass-burning aerosols by the IFS. It led to a maximum difference of 0.6 in the Aerosol Optical Depth (AOD) between the two formulations (Figure 4d). However, changes in AOD remained confined near the fires were negligible over Europe, despite the fact that the smoke plume from Quebec on reached Europe the 12 of July. Finally it is remarkable that changes in the biomass burning
emissions from GFAS have an impact over the whole 5 days of model integration.

### 3.3 Comparison with observations

The reduction in the GFAS emissions is consistent with what is expected from the new observation operator. It is more challenging to verify if the reductions are also improving the GFAS errors given the shortage of available measurements of fire emissions. When MODIS FRP observations are available (e.g. cloud free pixels) it is possible to assess the predictive skill of
persistence and of the model based on the FWI by using the FRP themselves as observations. Admittedly in this comparison only FRP in clear sky conditions are used even if persistence and the FWI model are instead applied to cloudy conditions i.e when FRP is not available. However cloud coverage only affects the capability of the satellite sensors to detect FRP. It is not accounted for in the FWI equation nor is it a component of the persistence. Therefore we assume it to be acceptable to extend the verification results to all sky conditions. Figure 5 shows the probability density function (pdf) of the differences between
the observed FRP and the two assumptions for the whole three month period. The comparison is performed using FRP of the previous day for the persistence and FRP*$O$ for the FWI experiment. Globally observed FRP values span a vast range with mean 0.14 $\mathrm{Wm^{-2}}$ and standard deviation 0.7 $\mathrm{Wm^{-2}}$. If persistence is assumed from the previous day the FRP is on average 40 % (mean error: 0.21 $\mathrm{Wm^{-2}}$) higher than observations. This highlights the limitation of using persistence. The bias in the FRP is reduced by the introduction of the new observation operator based on the FWI (mean error 0.16 $\mathrm{Wm^{-2}}$).
Aerosol Optical Depth (AOD) verification is difficult to perform in a statistically robust way as AOD observations are usually sparse in regions where fires are ignited and, as shown, changes in AOD are tend to be local. However figure 6 shows the daily variations of aerosol optical depth (at 550nm) observed at two stations in the proximity of the Hudson Bay. Both Pikle Lake and Chapais are part of the Aeronet network which provides globally distributed observations of spectral aerosol optical depth (AOD), inversion products, and precipitable water in diverse aerosol regimes Holben et al. (1998). Measurements
are compared with CAMS forecasts over the first 24 hours using fire emissions from the operational GFAS and the updated version with the FWI based model. For these stations, the impact of our proposed modification is mainly beneficial. For example, the overestimation in AOD on 12 July 2013 is nearly halved using the new observation operator. AOD forecasts were also improved at other AERONET stations in the region, such as Waskesiu (not shown). In total there were observations recorded at 90 AERONET stations covering North America for the period of interest and looking at global statistics there is a
small benefit in adopting the FWI method in GFAS (figure 7).




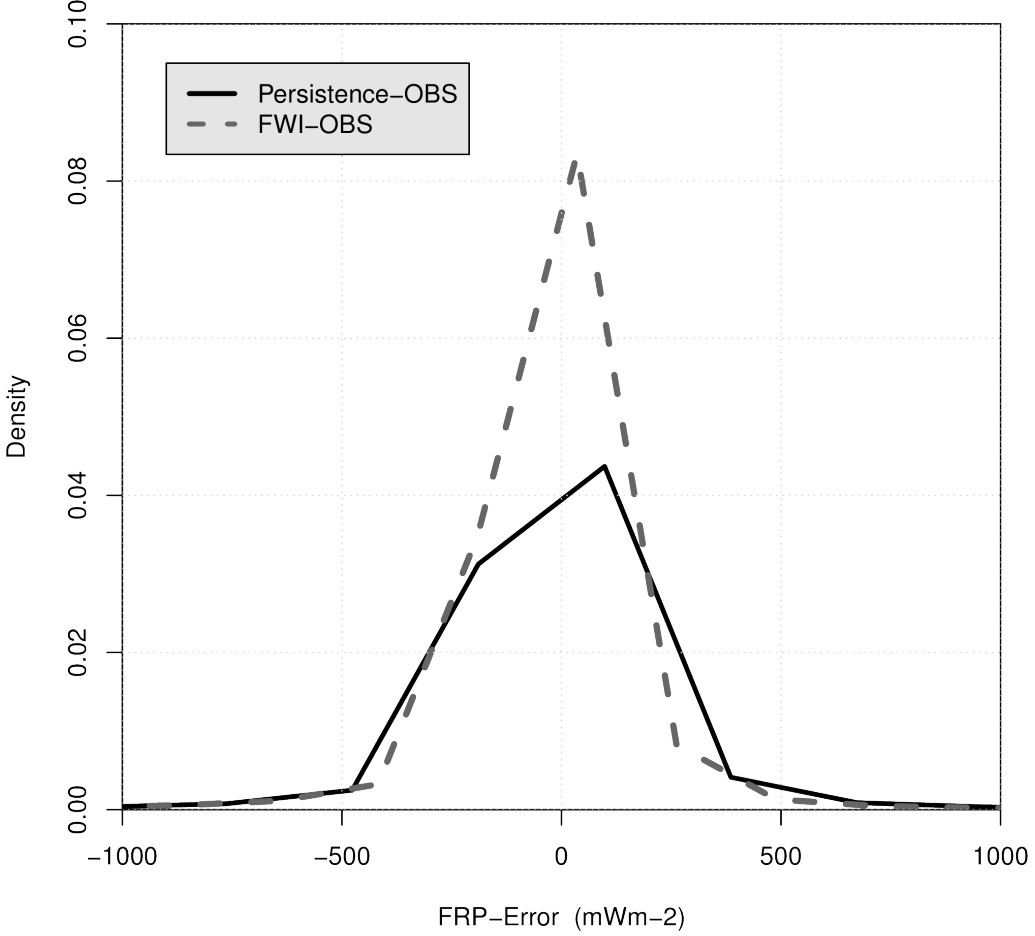

**Figure 5.** Global verification of FRP against observations. Probability density functions (pdf) of the observed FRP departure of MODIS compared to the two observation operator models relaying on the assumption of persistence and on the FWI.

## 4 Conclusions

In this paper we have proposed a new approach to improve the estimation of fire emissions from fire radiative power observations. The method, based on the likelihood of fire persistence evaluated using the Fire Weather Index has been implemented in the Global Fire Assimilation System to substitute the simple assumption of persistence in the case where there are missing

5 observations.

Results show that the new FWI-based observation operator is a more accurate predictor of FRP in case where there are missing observations. In particular, the overestimation of fire duration present in the operational GFAS is substantially reduced. Moreover the new fire-related emissions also have a positive impact on the aerosol forecast of the CAMS atmospheric composition system. The impact on the AOD forecasts during a large fire event in Canada in Summer 2013 is large close to the

10 fires and negligible elsewhere. One of the advantages of the new approach is that the concept of modeling the likelihood of fire





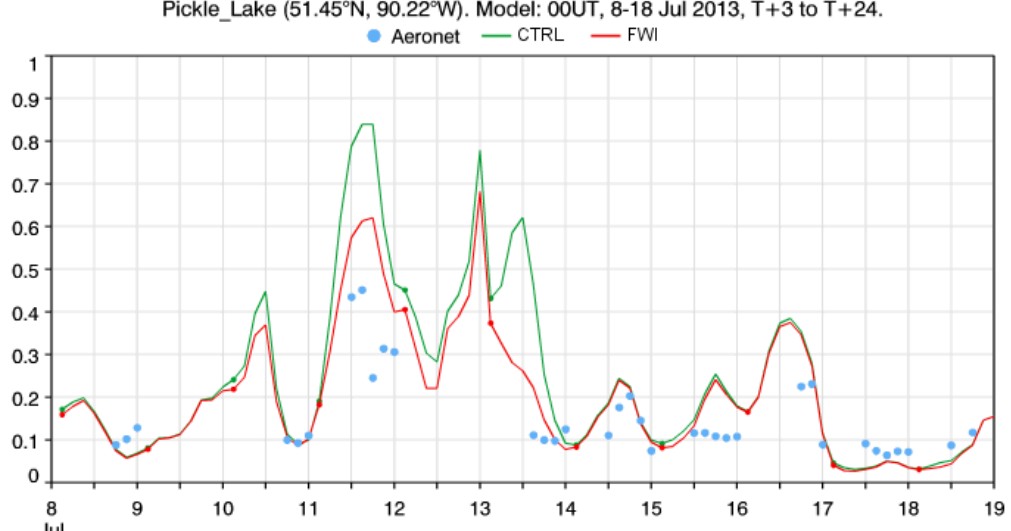

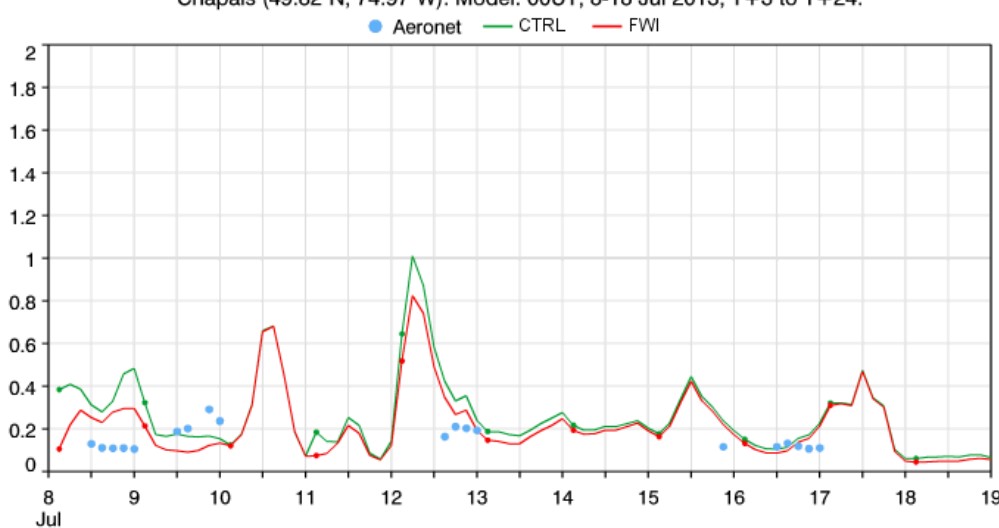

**Figure 6.** Local verification against observations. AOD at 550nm evolution at two Aeronet stations (Pickle lake and Chapais) which are in the proximity of the Hudson bay. Local measurements are compared to the two models simulations nearest grid-point. The model simulations are a concatenation of 24h forecasts. The starting of the forecast is marked with a dot.

persistence as a function of weather parameters can also be extended to modulate the contribution of fire emission during the model integration. This aspect has already been investigated in Di Giuseppe et al. (2017) where the FWI has been employed to forecast the time evolution of the fire emissions that GFAS provides to the CAMS.





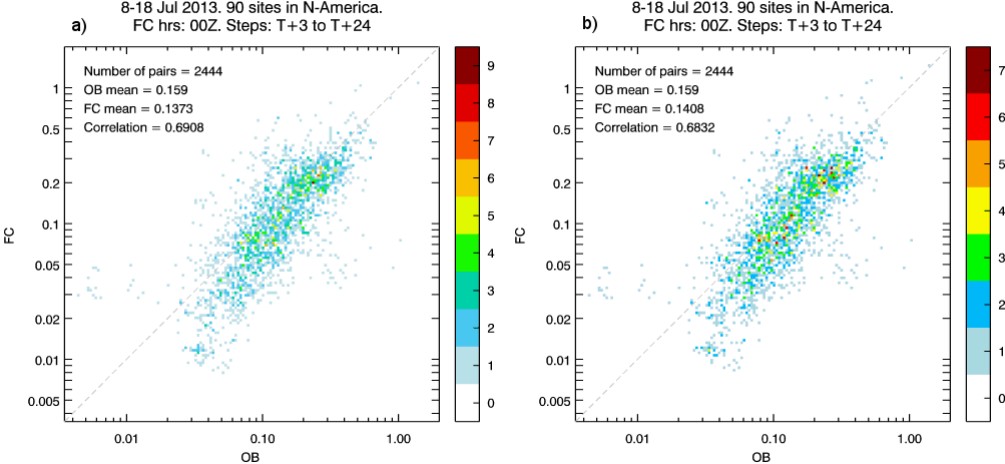

**Figure 7.** AOD at 550nm for the 90 Aeronet stations in North-America and the two CAMS simulations initialized with the operational GFAS (Panel a) and using the new FWI based model (Panel b). The model simulations are a concatenation of the first 24 hours forecasts.

*Data availability.* The Fire Radiative Power data produced by GFAS and the fire weather index (FWI) calculations using ecmwf model are available through the Copernicus data web portal at ECMWF http://apps.ecmwf.int/datasets



## Appendix A: Fire Weather index Calculation

A schematic illustration of the Fire Weather Index modeling components, their interactions, and the needed weather forcing is illustrated in figure A1

The first step in the FWI calculation is the evaluation of the fuel moisture content. In the FWI biomass fuel is approximated

5 with three non-interacting fuel layers characterized by their fast or slow response to the atmospheric forcing. The Fine Fuel Moisture Code (FFMC) provides a rating of the moisture in the litter and fine fuel occupy the first layer at the interface with the atmosphere. The Duff Moisture Code (DMC) characterizes the moisture of an intermediate layer which consists of loosely compacted organic material and finally the Drought Code (DC) calculates the moisture retained at the bottom the bottom where there is a deep, compact organic layer. After the diagnostic calculations of fuel moisture content in these three layers,

10 the model calculates fire behavior indexes in terms of rate of fire spread (Initial Spread Index, ISI) and fuel available for combustion (Build-up Index, BUI) taking into account the conditions of the previous day. Finally the FWI integrates these latter two quantities to produce a unit-less open scale index of general fire intensity potential.

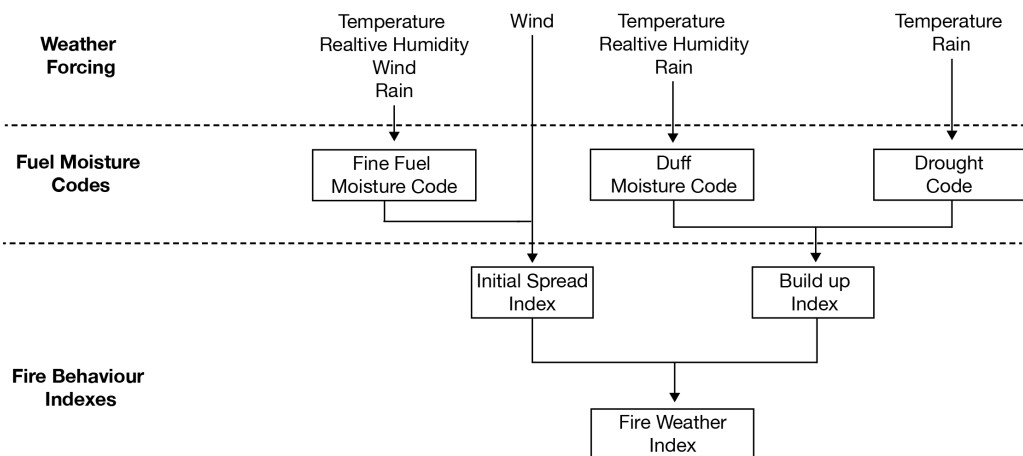

**Figure A1.** Schematic structure of the FWI



*Competing interests.* We declare no competing interests

*Acknowledgements.* This work was funded by the EU Project ANYWHERE (Contract 700099) and the Global Fire Contract 389730 between the Joint Research Centre and ECMWF. Angela Benedetti is thanked for the fruitful discussions concerning the interpretation of the MOPITT comparison. We would like to thank NASA and the MOPITT team for the ease of access to their data. Luke Jones is acknowledged for his

5   support in creating Figure 6. Rebecca Emerton and Sue Dunning kindly read the manuscript before submission.



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
