# Peer review of "Combining fire radiative power observations with the fire weather index improves the estimation of fire emissions"

_Atmospheric Chemistry and Physics, 2017_

## Referee Comment (RC1) · Anonymous Referee #1 · 3 Dec 2017

Overall, the manuscript was well written and concise. The figures and equations were explained well within the manuscript. The importance of the study was also adequately addressed. In general, the results support the conclusions; however, further information is needed (see specific comments) specifically related to the use of this model over various land cover types and in regions impacted by cloud cover.

Specific Comments:

* There is no mention of testing the model over different land cover/use types. Have the authors run any comparative tests? There was a brief mention of results over Africa that were not included. Could you include some additional results in the appendix - if

the manuscript length is an issue?

\* Page 2 Lines 11 – 20: Please include the nadir pixel resolutions for the SEVIRI and MODIS products for comparison. The authors mention the advantage of increasing the temporal frequency of observations using SEVIRI; however, this will lead to much coarser spatial resolutions, which will have implications on small fire mapping.

\* Page 2 line 21 – The authors should state that they are using MODIS FRP observations - for clarity after the SEVIRI paragraph.

\* Page 7 line 6 – Define extreme fire.

\* Page 7 Line 12 – "Using FWI…..up to 80%...." – could you include the average and std values too?

\* Page 9 line 13 – Have the authors tested their model during months when cloud cover is an issue?

Technical Comments

\* Page 2 line 28 – Change (Di Giuseppe et al., 2016) to Di Giuseppe et al. (2016).

\* Page 5 line 1 – Should "…is still large" read "…is still larger"?

\* Page 7 line 9 – Should 4 June read 4 July?

\* Page 7 line 8 – Change "on 2-6 July" to "between 2-6 July".

\* Page 7 line 14 – Change "i.e.." to i.e.

\* Page 9 line 4 – Change …"from Quebec on reached Europe the…" to "…from Quebec reached Europe on the…"

\* Page 9 line 24 – "Holben et al., 1998" should read (Holben et al., 1998)

\* Page 10 line 6 – change "…in case…" to in cases

---

## Referee Comment (RC2) · Anonymous Referee #2 · 11 Dec 2017

This is an important contribution to the literature on estimating emissions from fire radiative energy (FRE). The implications for existing methods of emissions estimates is timely, relevant, important, and perhaps most importantly, relatively straightforward to implement. The writing is very concise, perhaps in some places too much so. The authors only refer to fire radiative power (FRP), and not FRE, the integration of FRP over time, which is used to estimate biomass combustion, from which emissions estimates for various species are calculated. The use of a lagged FWI is an interesting choice, as several fire weather indices from the FWI family already have 'memory' from the previous day (fine fuel moisture code, duff moisture code and drought code). Also, FWI was developed for Canadian temperate and boreal forests, is it really appropriate to apply this method of inferring FRP to tropical and subtropical systems as the authors do (without good result)? It requires justification. Some terminology seems unclear, such as reference to "overestimation" (which implies some reliable reference) when comparing GFAS and MODIS data. Also, I'm not sure how "missing data" were identified, as no cloud masking algorithm is described. Is it possible that real zeros are being filled in with data using this method? This should be addressed in the text. Finally, is this a global analysis? The study area should be stated explicitly or if it's a global analysis it should say so.

Specific comments: P9L4: Why is this remarkable? Can you be more explicit? P9L7: Why is this the expectation? Couldn't it have just as easily underestimated emissions if peaks were missed? (Indeed, could this not still be happening if overpasses occur outside of peak burning hours?) P9L15: I'm not sure that "assumptions" is the right word here? Operator versus non-operator? These aren't assumptions, exactly, maybe "methods"? P9L16: Not sure about "experiment", as that's not really what this is, again "method" may be more appropriate. P9L18: This is great (40% result), it should be highlighted in the abstract.

---

## Author Comment (AC1) · 31 Jan 2018

**Answers to Anonymous Referee #1 "Combining fire radiative power observations with the fire weather index improves the estimation of fire emissions"**
**By Francesca Di Giuseppe et al.**

Overall, the manuscript was well written and concise. The figures and equations were explained well within the manuscript. The importance of the study was also adequately addressed. In general, the results support the conclusions; however, further information is needed (see specific comments) specifically related to the use of this model over various land cover types and in regions impacted by cloud cover.

We would like to thank the reviewer for the positive comments.

Specific Comments:
* There is no mention of testing the model over different land cover/use types. Have the authors run any comparative tests? There was a brief mention of results over Africa that were not included. Could you include some additional results in the appendix - if the manuscript length is an issue?

This is a very good point. The FWI was specifically designed to work for the boreal forecast of Canada and, despite it is used worldwide, its accuracy might be different under different ecosystems. This implies that modelling changes in FRP using changes in FWI might be more or less successful in different parts of the globe. We have tried to address this aspect using a land cover mask to divide among different ecosystems. For consistency we use the land cover mask that is implemented in GFAS to attribute the dry matter combustion rate to each grid point.

This land cover type is based on land cover classes which are derived from the dominant burning land cover type in GFED3.1 and additional organic soil and peat maps, (full details are given in Heil et al. 2010 which describes GFAS settings).

By using this land cover mask we have estimated the error between FRP and persistence and between FRP and the FWI model for all the points for which a valid FRP was observed. The 2d error density plots shown highlights that the FWI is very effective in reducing the errors when there is a overestimation of fire activity and works equally well in different land covers.

We have added the plot to the paper and and explanatory paragraph which reads

*In the new approach changes in FWI are used to predict changes in FRP. The accuracy of this assumption depends on how a good predictor is FWI of fire evolution. The FWI was developed to describe fire danger and behavior for the boreal forests of Canada and its accuracy might be smaller for different vegetations. To understand the expected reduction of error in different ecosystems figure \ref{fig:land_cover_class} partitions the results shown in figure \ref{fig:obs_comparison1} by land cover using GFAS classification. In GFAS the land cover classes are derived from the dominant burning land cover type in GFED3.1 and additional organic soil and peat maps, (full details are given in \citet{heil:10}). The density plots show a substantial reduction of the overestimation errors for all land cover types. One interesting aspect is that*

*positive biases are reduced more than negative ones. This behavior can be explained noticing that the distribution of FWI, by using FWI values only when fire events are observed, is conditionally sampled towards high values \citep{digiuseppe:17}. At extreme value, the FWI flattens out and increases in its value are limited. Instead an increase in precipitation and humidity can produce a sudden FWI decrease, which translate to large negative value for the modulation factor. Negative FWI increments are therefore larger than positive ones. This asymmetric behavior means that errors are mostly corrected for overestimation of FRP on missing observation locations.*

[Figure]

**Figure caption**

*Upper panel: Land cover class map based on dominant fire type in GFEDv3.1 and organic soil and peat maps. Gaps in land areas have been filled (see Kaiser et all (2012) for details). Lowe panels: 2-dimensional probability density functions (PDFs) of the observed FRP departure of MODIS compared to the two observation operator models relaying on the assumption of persistence and on the FWI. The PDF refers to different land covers a) Savanna (SA), b) Savanna with Organic Soil (SAOS),c) Agriculture (AG),d) Agriculture with Organic Soil (AGOS), e) Tropical Forest (TF), f) Peat (PEAT), g) Extra-Tropical Forest (EF), h) Extra-Tropical Forest with organic Soil (EFOS)*

* Page 2 Lines 11 – 20: Please include the nadir pixel resolutions for the SEVIRI and MODIS products for comparison.

We have added this information

"*SEVIRI has lower spatial resolution then MODIS; the observing pixel size at nadir is 3000m against the 1000m of MODIS for the channels that are used in the FRP calculations*."

The authors mention the advantage of increasing the temporal frequency of observations using SEVIRI; however, this will lead to much coarser spatial resolutions, which will have implications on small fire mapping.

Agree and we have added this information in the text

"*This broad spatial resolution makes small fire mappings challenging with the exclusive use of SEVIRI. Nevertheless, the high sampling frequency means fire observations can occur during brief cloud-free spells in otherwise mostly cloudy regions.*"

* Page 2 line 21 – The authors should state that they are using MODIS FRP observa- tions - for clarity after the SEVIRI paragraph.

Corrected

* Page 7 line 6 – Define extreme fire.
We agree - The use of "extreme" was therefore not appropriate and we have rewritten the sentence
Usually fires are classified as extreme (or mega-fire) if they are widely spread and rage out of control (scientific literature suggests that fires with an areal extent > 10 000 ha, are ecological disasters because they burn vast areas of land and are characterized by high intensities that are seemingly outside of observed historical ranges). However there is not a rigorous definition. n this sentence we did not refer to mega fires but describe a properties of the two fire cases considered. Moreover we want to showcase the capability of the new way of modelling FRP for average cases more than for extreme situation.

*To understand the impact of the new operator on the emissions we concentrate on the prediction of two fire events with different characteristics*

* Page 7 Line 12 – "Using FWI.....up to 80%...." – could you include the average and std values too?
The mean is 0.54 and the standard deviation is 0.30. We have added this information in the text

* Page 9 line 13 – Have the authors tested their model during months when cloud cover is an issue?

The model is tested globally and for locations where observations are missing due to cloud cover issues. So for example in figure 3 (Which is copied here for convenience), the green bars show the number of points where FRP observations were classified as missing in the MODIS dataset. The FRP predicted in these locations are different depending on the model applied. If persistence is applied than the total FRP budget is expressed as the blue line (top plot). Otherwise, if the FWI model is used the total FRP budget is expressed as the green line. Only missing FRP observations (mostly due to cloud cover) contribute to the differences between these two lines (red line).

[Figure]

Technical Comments

* Page 2 line 28 – Change (Di Giuseppe et al., 2016) to Di Giuseppe et al. (2016).
Corrected

* Page 5 line 1 – Should "is still large" read " is still larger"?
Corrected

* Page 7 line 9 – Should 4 June read 4 July?
Yes thanks for pointing this out

* Page 7 line 8 – Change "on 2-6 July" to "between 2-6 July".
Changed

* Page 7 line 14 – Change "i.e.." to i.e.
Corrected

* Page 9 line 4 – Change "from Quebec on reached Europe the " to " from Quebec reached Europe on the"
Changed

* Page 9 line 24 – "Holben et al., 1998" should read (Holben et al., 1998)
Changed

* Page 10 line 6 – change " in case " to in cases
Changed

---

## Author Comment (AC2) · 31 Jan 2018

**Answers to Anonymous Referee #2 "Combining fire radiative power observations with the fire weather index improves the estimation of fire emissions"**
**By Francesca Di Giuseppe et al.**

*This is an important contribution to the literature on estimating emissions from fire radiative energy (FRE). The implications for existing methods of emissions estimates is timely, relevant, important, and perhaps most importantly, relatively straightforward to implement. The writing is very concise, perhaps in some places too much so.*

We would like to thank the reviewer for appreciating this paper. We also believe is a first step toward taking a stronger modelling approach in the use of FRP observations to improve fire emissions. As he/she also points out this paper tries to prove a concept; that is possible to model FRP starting from previous day observed FRP and a simple function of weather conditions. The ease of implementation was a strong requirement as we wanted to test the approach in an operational setting. We obtained a limited (encouragingly) positive improvement. Probably by having a better model for the FRP, for example by considering other variables and not only the FWI as a predictor, there is scope for improve the results in the future.

*The authors only refer to fire radiative power (FRP), and not FRE, the integration of FRP over time, which is used to estimate biomass combustion, from which emissions estimates for various species are calculated. The use of a lagged FWI is an interesting choice, as several fire weather indices from the FWI family already have 'memory' from the previous day (fine fuel moisture code, duff moisture code and drought code).*

We model the FRP instead than the FRE as this is the variable that is mostly directly related to biomass burning consumption. Also the FWI is a measure of fire danger integrated over a day therefore changes in the FWI are more likely to be related to the integral over time of FRE.

*Also, FWI was developed for Canadian temperate and boreal forests, is it really appropriate to apply this method of inferring FRP to tropical and subtropical systems as the authors do (without good result)? It requires justification.*

The FWI was originally developed for the boreal forest however has been applied successfully in many other part of the planet. Showing where there could be large errors in modelling FRP changes as a function of FWI changes is a good suggestion. We have attempted to identify regional variations by using the land cover mask that is implemented in GFAS to attribute the dry matter combustion rate to each grid point. In GFAS the land cover classes are derived from the dominant burning land cover type in GFED3.1 and additional organic soil and peat maps (full details are given in Heil et al. 2010).

By using this land cover mask we have estimated the error between FRP and persistence and between FRP and the FWI model for all the points for which a valid FRP was observed. The 2d error density plots

shown highlights that the FWI is very effective in reducing the errors when there is a overestimation of fire activity and works equally well in different land covers.

We have added the plot to the paper and and explanatory paragraph which reads

*In the new approach changes in FWI are used to predict changes in FRP. The accuracy of this assumption depends on how a good predictor is FWI of fire evolution. The FWI was developed to describe fire danger and behavior for the boreal forests of Canada and its accuracy might be smaller for different vegetations. To understand the expected reduction of error in different ecosystems figure \ref{fig:land_cover_class} partitions the results shown in figure \ref{fig:obs_comparison1} by land cover using GFAS classification. In GFAS the land cover classes are derived from the dominant burning land cover type in GFED3.1 and additional organic soil and peat maps, (full details are given in \citet{heil:10}). The density plots show a substantial reduction of the overestimation errors for all land cover types. One interesting aspect is that positive biases are reduced more than negative ones. This behavior can be explained noticing that the distribution of FWI, by using FWI values only when fire events are observed, is conditionally sampled towards high values \citep{digiuseppe:17}. At extreme value, the FWI flattens out and increases in its value are limited. Instead an increase in precipitation and humidity can produce a sudden FWI decrease, which translate to large negative value for the modulation factor. Negative FWI increments are therefore larger than positive ones. This asymmetric behavior means that errors are mostly corrected for overestimation of FRP on missing observation locations.*

[Figure]

**Figure caption**

*Upper panel: Land cover class map based on dominant fire type in GFEDv3.1 and organic soil and peat maps. Gaps in land areas have been filled (see Kaiser et all (2012) for details). Lowe panels: 2-dimensional probability density functions (PDFs) of the observed FRP departure of MODIS compared to the two observation operator models relaying on the assumption of persistence and on the FWI. The PDF refers to different land covers a) Savanna (SA), b) Savanna with Organic Soil (SAOS),c) Agriculture (AG),d) Agriculture with Organic Soil (AGOS), e) Tropical Forest (TF), f) Peat (PEAT), g) Extra-Tropical Forest (EF), h) Extra-Tropical Forest with organic Soil (EFOS)*

*Some terminology seems unclear, such as reference to "overestimation" (which implies some reliable reference) when comparing GFAS and MODIS data.*

Our reference is what is observed in terms of FRP from MODIS. We then compare this value to the persistency assumption and the FWI based model.

*Also, I'm not sure how "missing data" were identified, as no cloud masking algorithm is described.*

GFAS uses the MOD14 product from MODIS collection 6. The MOD14 files contains a fire classification pixel which is used to process valid observations.
MOD14/MYD14 fire mask pixel classes

1. not processed (missing input data)
2. not processed (obsolete; not used since Collection 1)
3. not processed (other reason)
4. non-fire water pixel
5. cloud (land or water)
6. non-fire land pixel
7. unknown (land or water)
8. fire (low confidence, land or water)
9. fire (nominal confidence, land or water)
10. fire (high confidence, land or water)

We do not apply any masking algorithm ourself and just rely on MODIS classification. This clarification has been added in the text:

*The FRP ($\rho$) observations from MODIS used in this study are available at 1 km resolution from the MOD14 product of MODIS collection 6. This dataset also provides a pixel classification attribute to flag missing observations (mostly because of cloud cover) and low confidence measurements, which is used to disregard invalid observations. Valid observations are first area-weighted by the portion of satellite footprint included in the grid box area in order to provide mean FRP density on a grid box of 0.1 degree.*

*Is it possible that real zeros are being filled in with data using this method? This should be addressed in the text.*

No this is not possible. FRP =0 is a valid observation  and is therefore not processed neither with the FWI method nor with the persistence one. Thus, pixels for which, at  DAY -1:  FRP >0  (Observed "active "fire) and at  DAY 0:  FRP= NA (Missing observation) are  going to be filled by the modelling approach(es)

Instead, pixels for which, at DAY -1: FRP =0  (Observed "non-active "fire ) and at DAY 0: FRP=NA (Missing observation)  are **not** going to be filled by the modelling approach(es).

*Finally, is this a global analysis? The study area should be stated explicitly or if it's a global analysis it should say so.*

GFAS is a  global system and so is our analysis.  We have stated this clearly now in the abstract and in the introduction.

*The atmospheric composition analysis and forecast for the European   Copernicus Atmosphere Monitoring Services (CAMS) relies on biomass   burning fire emission estimates from the Global Fire Assimilation  System (GFAS). GFAS is a global system and converts fire radiative power (FRP) observations  from MODIS satellites into smoke constituents.*

*Specific comments: P9L4: Why is this remarkable? Can you be more explicit?*

The reviewer is right that this sentence needs more explanation. In IFS small changes in the initial conditions might not be visible in the forecast. If these changes  are "perceived" by the model as random noise they are just dumped as spurious oscillations in the model (there is a digital filter implemented in the IFS for this purpose ). The fact that changes produced by the introduction of the FWI persist through the forecast it means that they are "compatible" with the model dynamical state.
This has been now elaborated in the text.

*Finally it is remarkable that changes in the biomass burning emissions from GFAS which, are used to initialize IFS, have an impact over the whole 5 days of model integration, signifying that these changes are compatible with the model dynamical state. On the contrary random noise would be just dumped as spurious oscillations in the model by the  digital filter implemented in the IFS.*

*P9L7:Why is this the expectation? Couldn't it have just as easily underestimated emissions if peaks were missed? (Indeed, could this not still be happening if overpasses occur outside of peak burning hours?)*

This question is in  line with the previous question of the reviewer and highlights  that we need to be more careful in specifying our reference. In this paper our truth is provided by the FRP observed by MODIS.  If this is our reference and we compare *only* with valid observation  we expect the FWI model to provide a decrease in the predicted FRP and thus in the emissions. This is also visible in the plot included in the previous question. The decrease in predicted FRP is  is due to the fact that FWI negative increments are larger than positive ones. As a consequence we also expecte a decrease in emissions.

However this doesn't mean that GFAS in its standard configuration  is necessary underestimating emissions as to assess this we would need to compare to measurements of fire emission which are not available at the global scale. An hint that GFAS dataset might underestimate emission is provided by our figure 8 which shows an underestimation of aerosol optical depth  when compared to  90 Aeronet stations in North-America. However this result cannot be generalised. We have tried to be clearer rewording  the sentence as follows:

*The reduction in the GFAS emissions when the new FWI model is used is consistent with the fact that large FWI decrements are more likely than increments,  during fire burning events. Negative increments decrease modeled FRPs, when compared to persistence, and lower the predicted emissions.  It is more challenging to verify if the reductions are also improving the GFAS errors given the shortage of available measurements of fire emissions.*

*P9L15: I'm not sure that "assumptions" is the right word here? Operator versus non-operator? These aren't assumptions, exactly, maybe "methods"?*

Agree - changed text

*P9L16: Not sure about "experiment", as that's not really what this is, again "method" may be more appropriate.*

Agree - changed tex

*P9L18: This is great (40% result), it should be highlighted in the abstract*

We have reworded the abstract as follows:

*One   of the consequences of this assumption is an increase of fire   duration, which in turn translates into an increase of emissions estimated from fires if compared to what is available from observations.*